# Scandium-44: Diagnostic Feasibility in Tumor-Related Angiogenesis

**DOI:** 10.3390/ijms24087400

**Published:** 2023-04-17

**Authors:** György Trencsényi, Zita Képes

**Affiliations:** Division of Nuclear Medicine and Translational Imaging, Department of Medical Imaging, Faculty of Medicine, University of Debrecen, Nagyerdei St. 98, H-4032 Debrecen, Hungary; trencsenyi.gyorgy@med.unideb.hu

**Keywords:** aminobenzoyl–bombesin analogue (AMBA), aminopeptidase N (APN/CD13), angiogenesis, carcinogenesis, gastrin-releasing peptide receptor (GRPR), integrin, nitroimidazole, positron emission tomography (PET), Arg-Gly-Asp (RGD), scandium-44 (^44^Sc)

## Abstract

Angiogenesis-related cell-surface molecules, including integrins, aminopeptidase N, vascular endothelial growth factor, and gastrin-releasing peptide receptor (GRPR), play a crucial role in tumour formation. Radiolabelled imaging probes targeting angiogenic biomarkers serve as valuable vectors in tumour identification. Nowadays, there is a growing interest in novel radionuclides other than gallium-68 (^68^Ga) or copper-64 (^64^Cu) to establish selective radiotracers for the imaging of tumour-associated neo-angiogenesis. Given its ideal decay characteristics (E_β_^+^_average_: 632 KeV) and a half-life (T_1/2_ = 3.97 h) that is well matched to the pharmacokinetic profile of small molecules targeting angiogenesis, scandium-44 (^44^Sc) has gained meaningful attention as a promising radiometal for positron emission tomography (PET) imaging. More recently, intensive research has been centered around the investigation of ^44^Sc-labelled angiogenesis-directed radiopharmaceuticals. Previous studies dealt with the evaluation of ^44^Sc-appended a_v_b_3_ integrin–affine Arg-Gly-Asp (RGD) tripeptides, GRPR-selective aminobenzoyl–bombesin analogue (AMBA), and hypoxia-associated nitroimidazole derivatives in the identification of various cancers using experimental tumour models. Given the tumour-related hypoxia- and angiogenesis-targeting capability of these PET probes, ^44^Sc seems to be a strong competitor of the currently used positron emitters in radiotracer development. In this review, we summarize the preliminary preclinical achievements with ^44^Sc-labelled angiogenesis-specific molecular probes.

## 1. Introduction

### 1.1. Tumour-Related Angiogenesis

The formation of new blood vessels from pre-existing ones—denoted as angiogenesis—is required for tumour maintenance and development as well as for metastasis formation [1,2]. Angiogenesis is regulated by a delicate balance between a host of pro-angiogenic and antiangiogenic factors [3]. Hence, intensive attention has been placed on the assessment of cancer-associated neovascularization [4,5]. Producing various angiogenic factors, proteases, heparanase, digestive enzymes, and chemotactic and stimulatory factors, tumour cells exert a direct effect on capillary endothelial cells, induce the assembly of activated immune cells, and trigger the activity of stromal cells; thus, they create a pro-angiogenic niche in the tumour microenvironment [2,6].

Given that angiogenesis/neo-angiogenesis has crucial role in tumour development, propagation, and metastatic spread, angiogenic biomarkers are recorded as promising candidates for tumour imaging [7]. Vascular endothelial growth factor (VEGF), platelet-derived growth factor (PDGF), fibroblast growth factor-2 (FGF-2), ephrins, α_v_β_3_ integrins, aminopeptidase N (APN/CD13), fibronectin, nitroimidazole, matrix metalloproteinases, gastrin-releasing peptide receptor (GRPR), and E-selectin represent the most important angiogenic molecules [8]. The detection of angiogenesis-related markers makes timely tumour identification possible, and these biomolecules also serve as potential targets for anti-tumour treatment and patient follow-up [1,9]. Anti-angiogenic drugs can inhibit tumour expansion and decelerate, delay, or even inhibit tumour augmentation and metastasis formation [10,11]. Therefore, angiogenesis-directed molecular probes may lay the basis of personalised cancer diagnostics and treatment.

Non-invasive positron emission tomography (PET) or positron emission tomography/computed tomography (PET/CT) are regarded as the mainstay diagnostic tools in the detection of primary tumours and pertinent metastases. Furthermore, PET can be a useful means of angiogenesis-targeted drug development, authorization, and dose estimation [8,12]. Several PET radionuclides have been used in angiogenesis-directed diagnostical settings, including fluoride-18 (^18^F), gallium-68 (^68^Ga), copper-64 (^64^Cu), and even the gamma emitter indium-111 (^111^In). Although scandium (Sc) radioisotopes were already recognized as valuable radiometals for isotope diagnostic use in the late 1990s, they have been set aside for almost 20 years [13].

### 1.2. Scandium-44 (^44^Sc)

The following three radionuclides of Sc are well suited for either diagnostic or therapeutic applications: scandium-43 (^43^Sc), scandium-44 (^44^Sc), and scandium-47 (^47^Sc) [14]. Positron emitters ^43^Sc and ^44^Sc are relevant diagnostic radiometals, whereas β^−^-emitting ^47^Sc is the therapeutic sister with accompanying γ rays for imaging purposes [15,16].

There are two major ways to produce ^44^Sc. Via the proton irradiation of natural calcium or enriched ^44^Ca targets (through a ^44^Ca(p,n)^44^Sc nuclear reaction), the production of ^44^Sc can be easily obtained by applying low-energy cyclotrons [17,18,19]. Cyclotron-based production provides outstanding radiochemical yield (RCY) and radiochemical purity (RCP) for clinical applications [18,20]. Of note, ^44^Sc production via applying a cyclotron has been confirmed to be economically viable [17]. Although ^44^Sc can also be eluated from titanum-44/scandium-44 (^44^Ti/^44^Sc) generator systems, the radioactive-waste handling of long-lived ^44^Ti (62 ± 2 years) makes this approach challenging [21,22,23]. In addition, the demanding production of ^44^Ti implies another obstacle regarding the routine usage of ^44^Ti/^44^Sc generators [17,24]. Difficulties around the production of ^44^Sc may underpin why this radiometal came to the focus of exhaustive investigation as a potential labelling isotope so late.

^44^Sc, with physical properties of T_1/2_: 3.97 h (lately reported as 4.04 h), E_β_^+^_average_: 632 KeV, E_β_^+^_mean_: 0.63 MeV, and I:94.3%, has begun to emerge as a radiometal of particular interest in imaging, dosimetry, and treatment follow-up [20,25,26]. Positron emitter ^44^Sc also has a gamma co-emission of 1157 keV (99.9%), which may limit the radioactive dose that can be injected into patients [26]. However, this high-energy gamma radiation made the radiometal a precious candidate for the novel β + γ coincidence PET imaging [27]. Since the positron energy of the applied PET radionuclide is inversely proportional to the resolution of the reconstructed image, better spatial-image resolution and better quality could be obtained with ^44^Sc relative to other radiometals—for example, ^68^Ga (^68^Ga: E_β_^+^_max_: 1.9 MeV and E_β_^+^_mean_: 0.89 MeV vs. ^44^Sc: E_β_^+^_max_: 1474 KeV and E_β_^+^_mean_: 0.63 MeV) [28,29,30]. These advantageous physical characteristics together with its decay to nontoxic Ca make ^44^Sc widely feasible in PET and PET/CT imaging [13,31]. Given its small ionic radius, the chemistry of ^44^Sc is comparable to that of ^68^Ga; thus, ^44^Sc could be applied in several fields, including dosimetry investigations in theranostic settings and in trafficking ligands with longer pharmacokinetics and subsequent requirements for longer imaging times (even 24 h post administration) [32,33].

Prior studies have already dealt with the investigation of the feasibility of ^44^Sc in pre-therapeutic dosimetry [34,35]. Khawar et al. published a paper indicating that the pharmacokinetics of [^44^Sc]Sc-PSMA-617-based (PSMA (prostate-specific membrane antigen)) PET/CT imaging serves as a useful tool for the calculation of normal organ-absorbed doses and the maximum allowable activity in prostate-cancer patients prior to lutetium-177 (^177^Lu)-labelled PSMA-617 ([^177^Lu]Lu-PSMA-617) radiotherapy [36]. Pioneering clinical studies further strengthened the efficacy of ^44^Sc-labelled somatostatin analogue [^44^Sc]Sc-DOTATOC and [^44^Sc]Sc-PSMA-617 as valuable radiotracers in dosimetry settings [35]. Delayed acquisition granted by the longer half-life of ^44^Sc enables better pre-therapeutic dosimetry for [^177^Lu]Lu-PSMA-617 radiotherapy [34,37]. Clinical successes in the treatment of leukaemia, non-Hodgkin lymphoma, malignant melanoma, urinary bladder cancer, glioma, neuroendocrine tumours, and prostate cancer with alpha emitter bismuth-213 (^213^Bi)- and actinium-225 (^225^Ac)-linked radiopharmaceuticals project that these isotopes could also benefit from dosimetry estimations based on ^44^Sc [38]. Moreover, in the long run, other radionuclides, such as yttrium-90 (^90^Y), ioide-131 (^131^I), strontium-90 (^90^Sr), and phosphorus-32 (^32^P) may be potential candidates for theranostic applications. Although preclinical research findings are available on radiation-absorbed doses from ^47^Sc, the limited availability of the radiometal hampers its usage as a potential radiotherapeutic agent [39].

Similar to ^68^Ga, ^44^Sc is also able to form thermodynamically stable complexes with chelator DOTA (1,4,7,10-teraazacyclododecane-N,N′,N″,N″′-teraacetic acid) [40,41]. In addition to this, ^44^Sc seems superior to ^68^Ga in several facets. Given its longer half-life (T_1/2_ ^44^Sc: 3.97 h vs. T_1/2_ ^68^Ga: 68 min) and cyclotron-dependent extensive synthesis, ^44^Sc can be easily shipped to remote nuclear medical facilities. This contributes not only to the wider distribution of the radioisotope but also to the accomplishment of protein- or antibody-based PET studies with lengthened examination times [42]. Furthermore, radiopharmaceuticals with longer pharmacokinetic characteristics can also be proposed due to the longer half-life of the radiometal [17]. Moreover, the pharmacokinetic properties of ^44^Sc-appended imaging probes are largely comparable to those of the ^68^Ga-labelled counterparts [43]. The nearly four-hour half-life of ^44^Sc could be easily fitted to the pharmacokinetics of several targeting molecules, including peptides, antibodies, or their fragments, as well as oligonuclides, which makes facile radiopharmaceutical synthesis possible [13,17]. Additionally, long-lived ^44^Sc favours delayed imaging and the achievement of appropriate tumour-to-background (T/M) ratios. The time of imaging is of critical importance from the point of view of patient management and the scheduling of examinations. Therefore, the use of ^44^Sc-labelled radiopharmaceuticals would contribute to the establishment of a fluent workflow. 

^44^Sc-labelled PET radiopharmaceuticals could even be appropriate for intraoperative radio-guided surgery, including the detection of lymphatic metastases at later time points post injection [32]. Consequently, ^44^Sc-based PET probes could stand out as valuable imaging agents. ^47^Sc (T_1/2_ = 3.35 days)—the therapeutic match of ^44^Sc—emits a β^−^ radiation of a maximum energy of 0.600 MeV (31.6%) and 0.439 MeV (68.4%), which could be exploited in targeted radiotherapy [33,44]. Therefore, ^44^Sc/^47^Sc has exquisite potential as a radiotheranostic pair in PET diagnostics and in therapeutic settings [33,45]. Besides ^44^Sc and ^47^Sc, scandium-43 (^43^Sc) is another significant member of the group of Sc isotopes. Investigating the quantitative capabilities of ^43^Sc/^44^Sc and comparing it to ^18^F and ^68^Ga, Lima et al. published a paper indicating that the application of radiotracers labelled with the mixture of ^43^Sc/^44^Sc may be beneficial in clinical fields [46]. The findings of their phantom study proved that precise, quantitative PET/CT could be obtained by applying commercial PET systems [46]. Given that ^43^Sc does not have high-energy gamma emission, the use of ^43^Sc/^44^Sc could be especially important in terms of overcoming the limitation of ^44^Sc derived from gamma radiation. These favourable features propelled both preclinical and clinical investigations with ^44^Sc down new lines. Some experiments have been spawned to focus on the evaluation of the clinical feasibility of ^44^Sc-labelled molecules at a translational level as well as the comparison of ^44^Sc with other widely used radionuclides, such as ^68^Ga and ^64^Cu [47].

### 1.3. PET Radioisotopes Other Than Scandium-44 (^44^Sc)

The optimal short half-life (109.8 min) and the high positron abundance (β ≥ 97%) of ^18^F made it the most commonly used radioisotope for the labelling of PET biomolecules [48]. Positron emitter ^18^F possesses a relatively low positron energy (E_max_ =  0.635 MeV and E_mean_ =  0.250 MeV) and a short positron range within the tissue (maximum of 2.3 mm) [49,50]. These nuclear properties ensure ideal image quality and high spatial resolution obtained with ^18^F-labelled tracers [51]. The 109.8 min-long half-life is beneficial for synthesis procedures as well as for the performance of examinations of a few hours [51]. Owing to the short longevity, ^18^F-labelled radiopharmaceuticals can be safely administered without an increased risk of excess radiation. Moreover, another advantage is the facile cyclotron-based production, which provides immense quantities of the isotope at high specific activity [51]. Although a cyclotron is required to produce the radiometal, its decay characteristics make the distribution of ^18^F- and ^18^F-labelled radiotracers to distant nuclear medical laboratories without an on-site cyclotron possible. Even though ^18^F is well suited for the labelling of a wide range of small and medium-sized molecules, the radiolabelling of peptides with ^18^F is still cumbersome [51]. Hence, different isotopes have come into focus that address some of the difficulties associated with ^18^F-based radiotracers.

Up to now, the favourable physical characteristics of ^68^Ga and ^64^Cu made them attractive for the radiolabelling of a broad set of molecules. Currently, ^68^Ga-labelled PET radiopharmaceuticals are applied most frequently for labelling purposes of radiometal-based PET tracers. The short-lived positron emitter ^68^Ga (T_1/2_: 67.71 min ≈ 68 min, Eβ^+^_average_: 830 KeV, maximum β^+^ energy: 1.92 MeV, Iβ^+^: 89%, Eγ: 1077 KeV, Iγ: 3.2%) can be obtained from a germanium-68/gallium-68 (^68^Ge/^68^Ga)-generator system [52,53]. With such physical properties, decay characteristics, and easy accessibility from an on-site ^68^Ge/^68^Ga generator, ^68^Ga has emerged as the mainstay PET-imaging radiometal [32,52,54]. Since the radiosynthesis of peptides—complexed with different macrocyclic ligands—with ^68^Ga is easy to perform, the use of ^68^Ga is effective in imaging settings [55]. ^68^Ga-labelled DOTA-conjugated somatostatin analogues (SSTR), including DOTATOC, DOTATATE, and DOTANOC, meant a substantial step forward in the imaging of SSTR-positive neuroendocrine tumours [33].

The short half-life and related transport challenges, however, may limit the widespread implementation of ^68^Ga into clinical and preclinical settings. Owing to the short longevity of the radiometal, radiolabelling procedures are only possible in laboratories equipped with in-house ^68^Ge/^68^Ga-generator systems. In addition, due to its short half-life, ^68^Ga-labelling can be applied solely in case of small molecules and peptides featured with a rapid pharmacokinetic profile [17,56]. Furthermore, ^68^Ga-labelling is only suitable for PET examinations of relatively short duration. Because of the breakthrough of ^68^Ge and sorbent material, the purification and concentration of ^68^Ga is warranted following elution, which could also restrict its applicability in routine diagnostic usage [57]. Image noise associated with the elevated positron energy of the radionuclide constitutes another shortcoming of ^68^Ga imaging [32]. These facts render ^68^Ga-labelled radiotracers of limited attractiveness for centralized distribution.

The application of ^64^Cu may bridge the limitations derived from ^68^Ga-associated imaging. Given its longer half-life (T_1/2_: 12.7 h) and generator-independent production, the use of ^64^Cu (β^+^ emission, *E_average_* = 278 keV, abundance: 19%) seems to be economically more viable for PET imaging [58,59]. Due to its longer-lived nature, the easy transport to distant laboratories without an on-site cyclotron facilitates the integration of ^64^Cu-appended radiopharmaceuticals into diagnostics. Moreover, the 12.7 h half-life of the radiometal can be easily tailored to biomarker-targeting small and large molecules, peptides, antibodies, and nanomolecules with prolonged elimination kinetics [60]. Additionally, its longer half-life makes ^64^Cu-based radiochemical procedures facile [60]. The coordination chemistry of the radiometal enables complexation with various chelating agents, which further supports the frictionless performance of radiolabelling [60]. Therefore, ^64^Cu is extremely useful in the establishment of a broad set of radiotracers for diagnostic purposes. However, the use of ^64^Cu is not without shortcomings. Its concomitant β^−^ emission (β^−^ = 39.0%, *E* = 190.2 keV) and relatively low positron-branching ratio (17.6%) mean a meaningful extra radiation danger for patients [58,61,62]. Furthermore, in case of complexation, chelators must be customized to the complex redox chemistry of ^64^Cu [54].

Beyond ^44^Sc, the use of zirconium-89 (^89^Zr) has also been exhaustively investigated in PET imaging [63]. Given its favourable decay half-life (T_1/2_ = 3.3 days; 78.4 h), appropriate radiochemistry, and the accessibility of different chelating agents for complexation with the radioisotope, ^89^Zr is welcomed for the radiolabelling of PET-based imaging molecules [63]. The relatively long half-life of the radiometal could be easily adjusted to the pharmacokinetic profile of monoclonal antibodies (mABs); therefore, ^89^Zr seems to be well suited for the radiolabelling of mABs and for PET immunoimaging applications [63,64,65]. Several mAbs, including anti-human epidermal growth factor receptor 2 (HER2) trastuzumab, anti-epidermal growth factor receptor (EGFR) mAb cetuximab, anti-PSMA mAb J591, and anti-vascular endothelial growth factor (VEGF) bevacizumab, were successfully labelled with ^89^Zr for the PET imaging of breast cancer, squamous-cell carcinoma, prostate tumours, and ovarian tumours, respectively, at both preclinical and clinical levels [66,67,68,69].

Due to their decay characteristics (positron emission: E_max_ and E_average_, 897 keV and 396.9 keV, respectively), PET images obtained with ^89^Zr-labelled compounds are of adequate spatial resolution [70,71,72]. However, similar to ^44^Sc, ^89^Zr has spontaneous high-energy gamma radiation of 908.97 keV, which accounts for a major drawback of its usage [70]. Although the concomitant gamma radiation does not influence either the quality or the quantification of the PET scans, it needs to be taken into account when patient doses are determined [73,74]. Another potential disadvantage could be the limited availability of the radiometal [63].

Taking the abovementioned facts into account, ^44^Sc may therefore be a valuable substitute for the currently applied ^68^Ga and ^64^Cu in the establishment of peptide-based targeted PET radiopharmaceuticals. In this review, we provide a comprehensive overview of the role of ^44^Sc-labelled peptide-based radiopharmaceuticals in the molecular imaging of cancer-related angiogenesis (Figure 1). Table 1 summarises the preclinical studies with ^44^Sc-labelled PET radiotracers that selectively target angiogenic biomarkers. In Table 2 the most important physical characteristics of the discussed PET radioisotopes are displayed.

AMBA: aminobenzoyl–bombesin analogue; APN/CD13: aminopeptidase N; BBN: bombesin; GRPR: gastrin-releasing peptide receptor; NGR: asparagine–glycine–arginine; PET: positron emission tomography; RGD: Arg-Gly-Asp; ^44^Sc: scandium-44; ^47^Sc: scandium-47.

AAZTA: (1,4-bis(carboxymethyl)-6-[bis(carboxymethyl)]amino-6-methylperhydro-1,4-diazepine) AMBA: aminobenzoyl–bombesin analogue; AR42J: rat exocrine pancreatic tumour; BBN: bombesin; BN: bombesin; c(RGD)_2_: dimeric cyclic arginine–glycine–aspartic acid; c(RGDfK): cyclo(-Arg-Gly-Asp-d-Phe-Lys); DOTA: 1,4,7,10-teraazacyclododecane-N,N′,N″,N″′-teraacetic acid; ELISA: enzyme-linked immunosorbent assay; ^68^Ga: gallium-68; GRPR: gastrin-releasing peptide receptor; HaCaT: human immortal keratinocyte; ^125^I: iodine-125; KB: human epidermal carcinoma; NI: nitroimidazole; NODAGA: 1,4,7-triazacyclononane-1-glutaric acid-4,7-diacetic acid; NRP-1: neuropilin-1 co-receptor; PCa PC-3: prostate cancer; PET/CT: positron emission tomography/computed tomography; PET/MRI: positron emission tomography/magnetic resonance imaging; RGD: Arg-Gly-Asp; ^44^Sc: scandium-44; SCID: severe combined immunodeficient; 4T1: mouse breast cancer; U87MG: human glioblastoma; VEGF: vascular endothelial growth factor.

## 2. Integrin a_v_b_3_

A_v_β_3_ integrin has a central role in angiogenesis, neovascularisation, tumour growth, and related metastatic spread [80]. Built up by noncovalently assembled α and β transmembrane subunits, the a_v_b_3_ integrin heterodimer is able to adhere to a wide set of target molecules, including extracellular matrix (ECM) or soluble ligands and different cell-surface molecules [81,82]. The interaction of a_v_b_3_ integrin with ECM proteins, fibroblast growth factor-2 (FGF2), metalloproteinase MMP-2, activated platelet-derived growth factor (PDGF), insulin, and vascular endothelial growth factor (VEGF) receptors favours cellular adhesion, cell proliferation, migration, and invasiveness, as well as the inhibition of apoptotic processes [83,84].

Since a_v_b_3_ integrin is upregulated in a vast array of cancer cells, tumour endothelial cells, and new-born blood vessels, it is recorded as an impressive biomarker of angiogenesis in oncological tumour diagnostics [82]. In addition, the expression level of a_v_b_3_ integrin correlates well with angiogenic activity, which makes the receptor suitable for the trafficking of antiangiogenic therapy.

Arg-Gly-Asp (RGD) tripeptides specifically bind to integrin receptors [85]. The affinity of an RGD sequence containing peptides to a_v_b_3_ integrin has increased its attractiveness for the development of RGD-based PET and single-photon emission computed tomography (SPECT) radioindicators in the detection of tumour-associated angiogenesis. Intensive focus has been placed upon the evaluation of ^18^F- and ^68^Ga-labelled, integrin-targeted PET radiotracers such as [^18^F]F-galacto-RGD, [^18^F]F-fluciclatide, [^18^F]F-RGD-K5, [^18^F]F-FPPRGD2, [^18^F]F-alfatide, [^68^Ga]Ga-NOTA-RGD, and [^68^Ga]Ga-NOTA-PRGD2 [85,86,87,88,89,90,91,92]. More recently, ^44^Sc-based RGD radiopharmaceuticals have gained increasing interest for tumour-imaging scenarios.

Hernandez et al. used ^44^Sc that was obtained from a cyclotron by the proton irradiation of natural Ca metal targets for the labelling of α_v_β_3_ integrin–affine dimeric cyclic arginine–glycine–aspartic acid (cRGD)_2_ to assess the peptide selectivity of [^44^Sc]Sc-DOTA-c(RGD)_2_ at both the in vivo and in vitro levels [17]. The tumour-targeting capability, the specificity, and the binding potential of this radioisotope were assessed in vitro, in vivo, and ex vivo by performing competitive-cell binding assay, small-animal PET examinations, receptor-blocking studies, and organ-distribution experiments. For the in vivo evaluation of the α_v_β_3_-targeting capacity of [^44^Sc]Sc-DOTA-c(RGD)_2_, U87MG human glioblastoma-bearing female athymic nude mice were intravenously (iv.) administrated with 5.5–7.4 MBq of the radiotracer and then underwent microPET/microCT acquisition at different time points (0.5, 2, and 4 h post injection). The α_v_β_3_ integrin overexpression of the tumour cells and the vasculature of the U87MG experimental models is well established. In a bid to assess the α_v_β_3_ selectivity of cRGD, (cRGD)_2_, and DOTA-(cRGD)_2_ in vitro, a competitive cell-binding assay was performed using integrin-targeting radioligand ^125^I-echistatin. Moreover, in vivo and ex vivo receptor-blocking studies were further conducted with the simultaneous application of 3.7 MBq of [^44^Sc]Sc-DOTA-c(RGD)_2_ and 50 mg/kg (approximately 1 mg) of c(RGD)_2_ to confirm receptor selectivity. During the competitive assays, continuously elevating amounts of cRGD, (cRGD)_2_, and DOTA-(cRGD)_2_ were administered to 1 × 10^5^ U87MG cells that were pre-incubated with ^125^I-echistatin. As part of the ex vivo experiments, the weight and radioactivity of the harvested major tissues and organs, the blood, and the tumour were measured. The tracer-uptake values were obtained as mean %ID/g ± SD. Due to the significantly elevated tracer accumulation of the tumours, along with the negligible background activity, high-contrast PET images could be obtained at each investigated time point.

Quantitative in vivo analyses corresponding to the ex vivo data strengthened prominent [^44^Sc]Sc-DOTA-c(RGD)_2_ uptake in the tumourous lesions, with %ID/g values of 3.93 ± 1.19, 3.07 ± 1.17, and 3.00 ± 1.25 measured 0.5, 2, and 4 h after the injection, respectively. Similar to the PET data, insignificant ex vivo non-target activity was detected. Receptor-blocking studies showed the reduction of the radiotracer uptake of the tumours both in vivo and ex vivo. However, a (cRGD)_2_-derived decrease in the tracer concentration was recorded ex vivo in the background organs/tissues, which could be attributed to their physiological α_v_β_3_ expression.

Assessing the adhering affinities of cRGD, (cRGD)_2_, and DOTA-(cRGD)_2_, Hernandez et al. noted that the administration of the investigated peptides induced the concentration-dependent disposition of ^125^I-echistain. Although a binding competence 10 times stronger was depicted in the case of (cRGD)_2_ relative to its monomeric counterpart, DOTA complexation exerted no meaningful effect on the peptide-targeting affinity of (cRGD)_2_. Based on current literature data, Hernandez et al. were the first to propose a peptide-based PET radiopharmaceutical labelled with cyclotron-derived ^44^Sc. Therefore, they managed to strengthen the suitability of ^44^Sc for PET investigations in addition to the already-established radiometals. A summary of their study can be seen in Table 1.

In another study, Domnanich et al. performed an in vitro and in vivo comparison of DOTA- and NODAGA-conjugated cyclic RGD tripeptide radiolabelled with both ^44^Sc and ^68^Ga ([^44^Sc]Sc/[^68^Ga]Ga-DOTA/NODAGA-RGD) (as seen in Table 1) [76]. In this study they also assessed ^68^Ga- and ^44^Sc-labelled somatostatin analogue NOC ([Tyr3,1-NaI3]octreotide) complexed with the same chelators ([^44^Sc]Sc/[^68^Ga]Ga-DOTA/NODAGA-NOC). Besides in vitro stability examinations, biodistribution and in vivo PET-/CT-imaging studies of female athymic nude (CD-1 nude) mice bearing U87MG (human glioblastoma) and AR42J (rat exocrine pancreatic tumour) tumours were conducted. DOTA chelator-derivatized peptides (both RGD and NOC) labelled with either ^44^Sc or ^68^Ga showed high in vitro stability. Although ^68^Ga-labelled NODAGA-RGD and NODAGA-NOC were stable during the examination time, NODAGA-conjugated peptides radiolabelled with ^44^Sc did not present such results. The in vitro stability was evaluated in the presence of Fe^3+^ and Cu^2+^, as well. The instability of [^68^Ga]Ga-DOTA-RGD and [^68^Ga]Ga-DOTA-NOC was registered upon Cu2^+^ administration, whereas ^44^Sc-labelled molecules coordinated with NODAGA were less stable in the case of the addition of both metal cations. The ^44^Sc-labelled RGD peptides conjugated with either DOTA or NODAGA showed an identical tumour-distribution profile (4.88 ± 0.67% IA/g and 4.50 ± 0.77% IA/g for [^44^Sc]Sc-DOTA-RGD and [^44^Sc]Sc-NODAGAD-RGD, respectively, at 0.5 h post injection). Intriguingly, the hepatic uptake of [^44^Sc]Sc-DOTA-RGD was more elevated in comparison to the accumulation of [^44^Sc]Sc-NODAGAD-RGD 0.5 h after the tracer injections; however, at later time points similar liver activities were measured. As for other selected organs and tissues, Domnanich et al. experienced the same radiopharmaceutical accretion. The biodistribution profile of the ^44^Sc-labelled and ^68^Ga-labelled compounds was compared, as well. The liver showed increased uptake of the ^68^Ga-labelled DOTA-chelated peptides relative to the ^44^Sc-appended DOTA matches. This finding was in accordance with the in vivo PET data. Regarding other organs, the uptake of the ^68^Ga and ^44^Sc peptides was mostly identical. Since the ^44^Sc-appended molecules complexed with NODAGA demonstrated a largely identical tracer-accumulation pattern compared to the DOTA-chelated ones, NODAGA seems to be an attractive alternative chelator to more established DOTA for the development of ^44^Sc-labelled PET radiotracers [76]. 

In a study by Nagy et al., the applicability of mesocyclic chelating agent AAZTA (1,4-bis(carboxymethyl)-6-[bis(carboxymethyl)]amino-6-methylperhydro-1,4-diazepine) for conjugation with ^44^Sc-labelled PET radiometalated peptides was ratified [77]. The outstanding complex formation between Sc^III^ and AAZTA laid the basis for the proposal of ^44^Sc-labelled PET-imaging probes chelated by AAZTA. Tumour-naïve healthy control and 4T1 tumour-bearing BALB/c mice were used for the evaluation of the in vivo biodistribution of ^44^Sc^3+^, ^44^Sc(AAZTA)^−^, and ^44^Sc(CNAAZTA-c(RGDfK)) (c(RGDfK): cyclo(-Arg-Gly-Asp-d-Phe-Lys)). The choice of using 4T1 tumours for their experiments was reinforced by the integrin α_v_β_3_ overexpression of breast cancers [93]. Positron emission tomography/magnetic resonance imaging (PET/MRI) examinations were performed 30 and 90 min post administration of the free ^44^Sc^3+^, ^44^Sc(AAZTA)^−^ complex, and ^44^Sc-labelled CNAAZTA-conjugate. Discrete hepatic, pulmonary, and lienal uptake of ^44^Sc^3+^ was registered on the PET scans of the control group 90 min post injection. In contrast, ^44^Sc(AAZTA)^−^ showed no accretion in either the thoracic or the abdominal organs. Investigating the distribution pattern of ^44^Sc(CNAAZTA-c(RGDfK)), Nagy et al. revealed physiological accumulation in the liver, intestines, and urinary tract of the control BALB/c study mice. Based on the enhanced 4T1 tumour-tracer uptake, the tumour-homing capability of this newly produced α_v_β_3_ integrin-affine, ^44^Sc-labelled, AAZTA-conjugated imaging probe was confirmed. Comparing the radioactivity of the neoplasms with that of the non-target tissues and organs, a radiopharmaceutical concentration 25 times more elevated was encountered in the subcutaneously (*sc.*) growing 4T1 tumours than in the background tissues. Consequently, imaging with ^44^Sc(CNAAZTA-c(RGDfK)) ensured high T/M ratios, which enable precise differentiation between the neoplastic alterations and the surrounding healthy tissues. It can be concluded that the application of a chelator other than the one generally used (e.g., DOTA) has no influence on the imaging properties of ^44^Sc. Furthermore, accurate lesion detection and anatomical localization provided by the appropriate T/M ratios are of crucial significance in terms of PET-image reporting. These findings also support the feasibility of ^44^Sc-labelled PET probes in tumour diagnostics. The details of their study are presented in Table 1. In a recent study by Ghiani et al., the potential of an AAZTA-chelated PSMA inhibitor (B28110) radiolabelled with ^44^Sc ([^44^Sc]Sc-B28110) was confirmed for the PET imaging of prostate cancer [94]. For the accomplishment of their study, LNCaP tumour-bearing mice were generated by the sc. transplantation of 5 × 10^6^ PSMA-positive LNCaP (prostate cancer derived from a metastatic lymph-node lesion of a human with prostate cancer) tumour cells into the right-shoulder and the right-thigh region of CB17 SCID male mice. Among others, they investigated the in vivo and ex vivo biodistribution pattern and the tumour-targeting capability of ^44^Sc- and ^68^Ga-labelled B28110 ([^44^Sc]Sc-B28110 and [^68^Ga]Ga-B28110) by performing dynamic (0–90 min) and 20 min-long static (150 min post tracer injection) in vivo PET/MRI examinations in male CB17 SCID mice bearing PSMA-positive LNCaP tumours. Furthermore, the diagnostic feasibility of these novel AAZTA-based imaging probes was compared with that of the DOTA analogue PSMA-617, which contains the same binding sequence. Analysing the PET/MRI scans, increased radiotracer accumulation was registered in the tumorous regions 20 min after the tracer administration. Based on the time-activity curves, [^44^Sc]Sc-B28110 accumulation was statistically more enhanced in the LNCaP tumours in comparison to the other assessed tracers regarding all time points; however, the tumorous lesions could be clearly detected with the use of all three probes ([^44^Sc]Sc-B28110, [^68^Ga]Ga-B28110, and [^44^Sc]Sc-PSMA-617). Based on the ex vivo biodistribution data three hours post tracer application, [^44^Sc]Sc-B28110 and [^68^Ga]Ga-B28110 showed the highest concentration in the LNCaP tumours, although no remarkable difference was noted between the accumulation of these two compounds. The ex vivo results also revealed an uptake two times higher of the labelled B28110 derivatives relative to that of the [^44^Sc]Sc-PSMA-617. Therefore, the findings of Ghiani et al. also strengthened the suitability of AAZTA for the conjugation of ^44^Sc PET probes in diagnostic settings [94]. 

## 3. Bombesin- and Gastrin-Releasing Peptide Receptor (GRPR)

G-protein-coupled gastrin-releasing peptide receptors (GRPR) can be encountered in a diverse array of malignancies, including prostate, breast, colon, pancreatic, and pulmonary tumours [95,96,97,98,99,100,101,102]. Beyond cancer cells, tumour vessels show GRPR expression [103]. Gastrin-releasing peptide (GRP) and GRP analogues exert specific binding potential to GRPR in mammals [104,105,106]. Bombesin (BBN), which shares functional and structural similarities with mammalian GRP, also has high affinity towards GRPR [95,104]. Previous in vitro an in vivo studies have confirmed that both BBN and GRP possess angiogenic properties [107,108,109]. Besides modulating the morphology of neoplastic cells, tumour differentiation, and tumour proliferation, activated GRPR catalyses the overexpression of proangiogenic genes [110,111]. BBN-triggered GRPR induces PI3 downstream signalling pathways, including the phosphorylation of angiogenesis-related Akt (protein kinase B) [112,113,114]. Moreover, the suppression of the BBN/GRPR axis leads to the inhibition of tumour propagation and vascularization [113]. Taking the abovementioned research findings into account, radiolabelled GRPR-targeting BBN or its analogues serve as valuable vectors in receptor-positive lesion detection.

Aminobenzoyl–bombesin analogue (4-aminobenzoyl-Q-W-A-V-G-H-L-M-NH_2_, AMBA) is regarded as a highly valuable imaging agent in the nuclear medical diagnostics of GRPR-upregulated tumours [115]. A former preclinical study proved the GRPR-binding ability of ^18^F- and ^64^Cu-labelled AMBA derivatives [116]. Furthermore, the targeting potential of DOTA-conjugated AMBA labelled with ^68^Ga or ^177^Lu was also reported at the preclinical level [115,117]. 

Investigating the GRPR selectivity of [^44^Sc]Sc-NODAGA-AMBA, Kálmán-Szabó et al. performed in vitro receptor binding and in vivo examinations using a GRPRpos. PCa PC-3 (prostate cancer) xenograft (displayed in Table 1) [75]. The application of PC-3 cell lines was suitable for these studies, as previous research had confirmed their GRPR positivity [111,118]. The imaging characteristics of [^44^Sc]Sc-NODAGA-AMBA were also compared with the ^68^Ga-labelled match ([^68^Ga]Ga-NODAGA-AMBA). In addition, the pharmacokinetic profile of [^44^Sc]Sc-NODAGA-AMBA and its in vivo and in vitro stability were determined. Based on the in vitro serum-stability measurements, the RCP of the ^44^Sc-appended compound was over 98% at both the 15 and 90 min time points, whereas the RCP of the ^68^Ga-tagged AMBA derivative showed a gradual decrease from 92% to 85% between 15 and 90 min. To authenticate the receptor-binding affinity of [^44^Sc]Sc-NODAGA-AMBA and [^68^Ga]Ga-NODAGA-AMBA, in vitro cellular-uptake studies were executed by applying GRPR-overexpressing human PCa PC-3 and receptor-negative human immortal keratinocyte HaCaT cell lines. Followed by a 60 or 120 min-long incubation with 0.37 MBq of [^44^Sc]Sc-NODAGA-AMBA or [^68^Ga]Ga-NODAGA-AMBA, the samples were measured for radioactivity in a gamma counter to provide uptake values in %ID/million cell units. Receptor-upregulated PC-3 cells presented notably higher (25 times) accumulation of both [^44^Sc]Sc-NODAGA-AMBA and [^68^Ga]Ga-NODAGA-AMBA than receptor-negative HaCaT cells (*p* < 0.01). Although from statistical point of view no considerable difference was encountered between the accumulation of the ^44^Sc-appended and the ^68^Ga-labelled radiotracers in the tumour cells, they expressed relatively more elevated [^44^Sc]Sc-NODAGA-AMBA uptake compared to [^68^Ga]Ga-NODAGA-AMBA (*p* < 0.05). Furthermore, in vitro blocking investigations performed with 200 nM BBN as blocking agent revealed that the tracer uptake of the PC-3 cells was remarkably diminished, whereas the radioactivity of the control cells remained unchanged. Analytical radio-HPLC-based in vivo serum-stability calculations conducted in healthy mice proved outstanding metabolic stability in the case of both probes. According to the pharmacokinetic investigations—similar to [^68^Ga]Ga-NODAGA-AMBA—[^44^Sc]Sc-NODAGA-AMBA possessed a circulatory half-life shorter than 30 min.

In a bid to conduct in vivo and ex vivo measurements, 12-week-old, severe combined immunodeficient (SCID) CB17 mice bearing PC-3 prostate cancer in their left-shoulder area and healthy control mice were examined. For the assessment of the biodistribution pattern of the probes, 60 and 120 min post injections of 11.3 ± 1.4 MBq of [^44^Sc]Sc-NODAGA-AMBA or [^68^Ga]Ga-NODAGA-AMBA static PET acquisition were conducted with the application of a miniPET device (University of Debrecen, Faculty of Medicine, Department of Medical Imaging, Division of Nuclear Medicine and Translational Imaging). Standardized uptake values (SUV) and T/M ratios were registered to quantify the radiopharmaceutical uptake. Both qualitative analyses and quantitative SUV data confirmed the GRPR-targeting adequacy of the investigated radiolabelled AMBA derivatives. Although 60 min post administration of [^44^Sc]Sc-NODAGA-AMBA higher SUV_mean_ (0.90 ± 0.17), SUV_max_ (1.54 ± 0.18), T/M SUV_mean_ (6.16 ± 1.24), and T/M SUV_max_ (6.71 ± 1.08) values of the sc. growing PC-3 tumours were registered compared to the ^68^Ga-labelled counterpart, this was not statistically significant (*p* ≤ 0.05). Moreover, 120 min after the injection of the radiotracers, decreased non-target activity was observed, which led to excellent T/M ratios.

Ex vivo biodistribution studies of both the healthy control and the tumour-bearing mice were accomplished 30, 60, 120, and 180 min after the administration of 11.3 ± 1.4 MBq of [^44^Sc]Sc-NODAGA-AMBA or [^68^Ga]Ga-NODAGA-AMBA. The radioactivity of the explanted tissues and organs was registered with a calibrated gamma counter, and the counts-per-minute (CPM) values of all samples were converted to the percentage of administered dose per gram of tissue. The radiotracer concentration was presented as %ID/g. The biodistribution pattern of the control small animals displayed lower accumulation of [^44^Sc]Sc-NODAGA-AMBA relative to [^68^Ga]Ga-NODAGA-AMBA; however, this difference was not statistically significant (*p* < 0.05). After analysing the tracer uptake of the organs separately, the blood, liver, spleen, small and large intestines, stomach, heart, and lungs were depicted with moderate radioactivity, whereas the kidneys, urine, adrenal glands, and pancreas demonstrated significant tracer accretion. The outstanding urinary activity could have been due to the renal method of elimination or the physiological presence of GRPR in the urogenital smooth muscle [119]. The natural appearance of GRPR in the pancreas and in the adrenal glands underlies the remarkable radiotracer uptake of these organs [105,120]. Discrete receptor expression of the neuroendocrine gastrointestinal cells and the pulmonary existence of the GRPR gene may explain tracer accumulation in the lungs and in the intestines [121,122]. In line with the in vivo findings, the [^44^Sc]Sc-NODAGA-AMBA uptake of the PC-3 tumours was more prominent in comparison to the ^68^Ga-labelled derivative.

Receptor-supressing in vivo and ex vivo studies were carried out with the injection of 15 mg/kg of BBN into the tumorous study mice 30 min before radiotracer application to further ratify the GRPR selectivity of the labelled BBN analogues. Correspondingly to the ex vivo blocking figures, both qualitative and quantitative in vivo analyses revealed considerably mitigated tracer uptakes in the PC-3 tumours 60 and 120 min post injection.

Given the GRPR selectivity of [^44^Sc]Sc-NODAGA-AMBA established by Kálmán-Szabó et al., along with the better imaging characteristics of ^44^Sc over ^68^Ga, ^44^Sc-labelled BBN-based radiotracers, will hopefully be embraced in the diagnostic armamentarium of GRPRpos. tumours.

Another investigation, executed by Koumarianou et al., focused on the comparison of ^44^Sc- and ^68^Ga-labelled BBN analogues in in vitro and in vivo biological medium [33]. In a bid to evaluate the potential impact of ^44^Sc on the binding ability of BBN analogue to GRPR, GRPR-affine DOTA-conjugated BBN analogue was labelled with ^68^Ga and ^44^Sc, and the in vitro receptor-adhering capacity, the ex vivo organ distribution, and the in vivo behaviour of both ^44^Sc-DOTA-BN[2-14]NH_2_ and ^68^Ga-DOTA-BN[2-14]NH_2_ were assessed. Table 1 shows the major points of their research.

DOTA-BN[2-14]NH_2_ complexed with ^nat^Ga and ^nat^Sc were used as non-radioactive derivatives to study in vitro GRPR-binding competence in competition with radiolabelled BBN analogue [^125^I-Tyr^4^]-BN in the human androgen-independent prostate-cancer PC-3 cell line. Thereafter, organ distribution was investigated in healthy male Sprague-Dawley rats one and two hours after the intravenous (iv.) administration of 11 and 3 MBq ^68^Ga-DOTA-BN[2-14]NH_2_ and ^44^Sc-DOTA-BN[2-14]NH_2_, respectively, via the jugular vein. As part of the biodistribution studies, blocking experiments were also performed with the iv. application of native BBN (100 µg/100 µL) 15 min before the radiotracer injection. The organ-distribution pattern of the control cohort—solely receiving the radiolabelled analogue—and that of the blocked group was compared. Androgen-independent Dunning R-3327-AT-1 prostate-cancer tumour-bearing small animals were generated by the subcutaneous (sc.) transplantation of GRPR-overexpressing R-3327-AT-1 cells (≈0.4 mL, 10^4^ cells/µL) into the dorsal aspect of the hind foot of male Copenhagen rats [33,123]. Dynamic microPET acquisition was executed approximately 10–14 days post tumour induction by applying 30–50 MBq of either the ^68^Ga- or the ^44^Sc-appended imaging probes. ^nat^Ga-DOTA-BN[2-14]NH_2_ showed higher specificity towards GRPR receptors than the ^nat^Sc-complexed counterpart. The pancreas was presented with considerable uptake using both labelled compounds with ID/g values of 0.64 ± 0.00% and 0.58 ± 0.05% for ^68^Ga-DOTA-BN[2-14]NH2 1 and 2 h post injection, respectively, whereas the subsequent data were obtained in the case of the ^44^Sc-labelled probe: 2.67 ± 0.53% ID/g and 1.51 ± 1.19% ID/g one and two hours after tracer administration, respectively. The physiological pancreatic GRPR expression could explain these results. Upon visual assessment of the in vivo PET images, the peripheral tumour areas demonstrated more enhanced tracer uptake compared to the central neoplastic regions. The heterogeneity of the receptor expression within the tumour and the unevenness regarding the radiopharmaceutical binding ability of GRPR could underpin the difference between the tracer accretion of the external and inner tumour regions. Followed by prompt radiotracer uptake post injection, Koumarinaou et al. noted a gradual decline in the accumulation kinetics of the ^68^Ga-labelled compound throughout the entire examination period, whereas they observed a stable albeit lower tracer uptake in the case of the ^44^Sc-appended match. Of note, the relative tumour radioactivity-based kinetics (normalized to the testis as a reference tissue) did not reveal any meaningful differences between the two tracers. Overall, the GRPR selectivity of ^68^Ga and ^44^Sc differed. However, given the similar distribution pattern, uptake, and excretion times of both the ^68^Ga- and the ^44^Sc-labelled radiopharmaceuticals, as well as their comparable accumulation in neoplastic tissues, these radiometals seemed equally effective in the detection of GRPRpos. tumours [33]. 

## 4. Hypoxia-Associated 2-Nitroimidazole (NI) Derivatives

Beyond the above-mentioned radiopharmaceuticals, hypoxia-associated PET tracers are also embraced in tumour diagnostics [124,125,126]. Since cancer-related neoangiogenesis, induced by oxygen insufficiency, leads to morphologically and functionally impaired blood-vessel development, which directly affects the efficiency of anti-tumour treatment, the early identification of hypoxic tumorous regions is of paramount importance [127,128,129]. In this respect, hypoxia-directed imaging probes could be potential weapons in the timely diagnostic assessment of tumour-related hypoxia and the achievement of therapeutic successes.

Given the fact that hypoxia-associated 2-nitroimidazole (NI) derivatives are intracellularly trapped under hypoxic conditions while they are reoxidised and excreted from normoxic cells, by applying NI-based imaging probes, healthy and hypoxic tumorous cells could be definitively differentiated from each other [130]. Therefore, NI compounds labelled with radioisotopes seem to be promising indicators of tumour-associated hypoxia. Previous successes were attained in clinical trials by applying ^18^F-labelled NI-derivatives such as [^18^F]F-fluoromisonidazole ([^18^F]F-FMISO) and [^18^F]F-fluoroazomycin arabinoside ([^18^F]F-FAZA) in hypoxia imaging [131,132]. Barthel et al., Yang et al., and Ziemer et al. proposed [^18^F]F-fluoroetanidazole ([^18^F]F-FETA), [^18^F]F-fluoroerythronitroimidazole ([^18^F]F-FETNIM), and [^18^F]F-[2-(2-nitro-1[H]-imidazol-1-yl)-*N*-(2,2,3,3,3-pentafluoropropyl)-acetamide] ([^18^F]F-EF5), respectively, for the non-invasive detection of cancer-related hypoxia [124,125,126]. However, the low T/M ratios generated by the prolonged elimination of these lipophilic tracers may hamper their widespread application in diagnostic fields. In 2011, Hoigebazar et al. verified the feasibility of ^68^Ga-labelled DO3AM-NI in hypoxia PET imaging [133]. 

Later in 2022, Szücs et al. published the synthesis of ^44^Sc-appended hypoxia-specific DO3AM-NI ([^44^Sc]Sc-DO3AM-NI) and its comparison to the ^68^Ga-labelled diagnostic match [78]. To study the in vivo and the ex vivo biodistribution of the ^44^Sc and ^68^Ga hypoxia PET probes, healthy and CB17 SCID adult male mice bearing KB (human epidermal carcinoma) tumours in their left-shoulder area were applied (presented in Table 1). In vivo PET/MRI studies were conducted 13 ± 1 days post implantation of 5 × 10^6^ KB tumour cells at an average tumour bulk of 110 mm^3^. Both the control and the tumourous study mice were iv. administered with 8.42 ± 0.38 MBq of [^44^Sc]Sc-DO3AM-NI and [^68^Ga]Ga-DO3AM-NI. Then, whole-body T1-weighted MRI and 20 min-long static whole-body PET acquisition was performed 90 and 240 min post tracer application. SUV and T/M ratios (ratios of tumour to skeletal muscles of the right-shoulder region) were defined as quantitative PET parameters. Ninety minutes and 4 h (240 min) post injection, organ-tissue specimens were harvested, weighted wet, and measured for radiopharmaceutical uptake by applying a gamma counter to obtain ex vivo data. Values were provided as mean %ID/g ± SD. Szücs et al. managed to label the NI derivative DO3AM-NI with ^44^Sc with a high-labelling yield and RCP. Except for the kidneys and the urinary bladder, the organs of the abdomen and thorax displayed moderate tracer accretion upon visual assessment of the PET scans. The high uptake of both probes in the kidney underpinned its important role in radiopharmaceutical elimination. Intriguingly, enhanced hepatic [^68^Ga]Ga-DO3AM-NI concentration was visually depicted 90 min post injection. The ^44^Sc-labelled NI-based counterpart accumulated rapidly in the sc. developing KB tumours and produced satisfactory images at 90 min and 240 min post tracer injection (SUV_mean_: 1.46 ± 0.32, 0.67 ± 0.03 at 90 and 240 min time points, respectively; SUV_max_: 2.35 ± 0.47 and 1.55 ± 0.28 at 90 and 240 min time points, respectively). Although the quantitative tumour uptakes of [^44^Sc]Sc-DO3AM-NI and [^68^Ga]Ga-DO3AM-NI were comparable, the liver, spleen, kidneys, intestines, lung, heart, and brain were depicted with higher [^68^Ga]Ga-DO3AM-NI uptake compared to the ^44^Sc counterpart 90 and 240 min post injection. More elevated accretion of the ^68^Ga-linked tracer in the non-target organs led to T/M ratios 10–15 times lower relative to the ^44^Sc-labelled derivative. In the case of [^68^Ga]Ga-DO3AM-NI, T/M SUV_mean_ and T/M SUV_max_ values of 85.66 ± 6.06 and 41.72 ± 13.61 were experienced, respectively, whereas for [^44^Sc]Sc-DO3AM-NI the following parameters were defined: T/M SUV_mean_: 182.67 ± 23.50 and T/M SUV_max_: 129.68 ± 26.11 4 h after the injection. Hence, outstanding lesion identification, which could be derived from the suitable T/M ratios of the ^44^Sc-labelled derivative, warrants a place for ^44^Sc in the targeted PET imaging of hypoxia-related tumour diagnostics. Unlike these findings, previous studies dealing with ^18^F, iodine-131 (^131^I), technetium-99m (^99m^Tc), and ^64^Cu-appended NI imaging probes reported substantial tracer accumulation in the abdominal organs, mainly in the intestines, along with hepatic metabolization, which could lead to difficulties regarding the detection of abdominal malignancies [134,135,136,137]. 

In line with the in vivo findings, the ex vivo figures revealed that the KB tumour uptake of the two investigated probes did not differ significantly between the two assessed time points; however, the accumulation of the ^44^Sc pendant was more prominent in comparison to the ^68^Ga-labelled compound both 90 and 240 min post injection. In agreement with the in vivo results, considerably higher T/M ratios were registered in connection with [^44^Sc]Sc-DO3AM-NI than [^68^Ga]Ga-DO3AM-NI (*p* ≤ 0.01). In addition, the ex vivo organ-distribution pattern also confirmed notably lower [^44^Sc]Sc-DO3AM-NI uptake in the evaluated organs compared to the ^68^Ga-labelled sister compound (*p* ≤ 0.01). The biodistribution results of Hoigebazar et al. and Seelam et al. regarding NI compounds labelled with ^68^Ga were in accordance with those of Szücs et al. [138,139]. In addition to the urinary method of excretion, they registered discrete radioactivity in the abdominal and thoracic organs [138,139]. As for the visualization of the PET scans, due to prompt urinary clearance and favourable T/M ratios, [^44^Sc]Sc-DO3AM-NI seemed to be superior to the ^99m^Tc-, ^64^Cu-, ^18^F-, ^131^I-, and ^68^Ga-linked derivatives. Based on the exceptional tumour-homing capability, appropriate contrast and favourable quantitative parameters of the ^44^Sc-appended molecular agent [^44^Sc]Sc-DO3AM-NI could be a highly valuable hypoxia-targeting PET probe in the diagnostic algorithm of tumour identification.

## 5. Vascular Endothelial Growth Factor Receptor (VEGFR)

VEGFs, including VEGF-A, -B, -C, -D, and -E, and placenta growth factor (PlGF) also have a pivotal role in blood-vessel formation [140,141]. The interplay between VEGF-A, VEGFR-2, and neuropilin-1 (NRP-1) co-receptor constitutes a crucial step in angiogenesis [142]. In addition, NRP-1 is presented in different cancer-cell lines and primary malignancies, including lung, renal, and breast cancer, as well as glioblastoma multiforme [143,144,145,146]. Consequently, biomolecules designed to target VEGF, its receptors, or NRP-1 have gained increasing attention in oncological research [147]. 

Although no previous in vivo experiments that investigated ^44^Sc-labelled VEGF-VEGFR-NRP-1-directed radiopharmaceuticals have been published so far, Masłowska et al. examined the potential diagnostic and therapeutic feasibility of such molecules under in vitro circumstances [79].

With the aim of inhibiting the formation of the vascular endothelial growth factor/neuropilin-1 co-receptor complex VEGF-A_165_/NRP-1, Masłowska et al. developed NRP-1-targeting radiotracers [79]. In their study, three NRP-1-affine molecules were labelled with ^44^Sc: Ala-Thr-Trp-Leu-Pro-Pro-Arg (A7R) heptapeptide, branched peptidomimetic Lys(hArg)-Dab-Pro-Arg (K4R), and the retro–inverso isomer of A7R (dArg-dPro-dPro-dLeu-dTrp-dThr-dAla) named ^D^R7A. For verification of their NRP-1 selectivity, a competitive NRP-1-binding inhibition assay was performed using enzyme-linked immunosorbent assay (ELISA). After the addition of recombinant human NRP-1 (100 µL; 200 ng/well) to a 96-well plate, 50 µL of the solution of the cold reference molecules of the assessed tracers and 50 µL (400 ng/mL) of human biotinylated VEGF-A_165_ ((bt)-VEGF-A_165_) were administered. Wells treated with (bt)-VEGF-A_165_ were the positive controls, whereas wells without NRP-1 were the negative controls. The IC_50_ values of the following cold biomolecules—Sc-DOTA-Ahx-A7R, Lys(DOTA-Sc)-A7R, and Lys(hArg)-Dab(Ahx-DOTA-Sc)-Pro-Arg—indicated high affinity towards NRP-1, meaning that their radiolabelled counterparts ([^44^Sc]Sc-DOTA-Ahx-A7R, [^44^Sc]Lys(DOTA-Sc)-A7R, and [^44^Sc]Sc-DOTA-Ahx-K4R) could be effectively used to prevent the development of VEGF-A_165_/NRP-1 complex. Hence, they could widen the existing scale of the diagnostic and therapeutic probes applied in cancer-related angiogenesis. In contrast, retro–inverso peptide ^D^R7A-based compounds exerted low NRP-1 specificity. However, due to the inadequate human-serum stability of the evaluated agents, future studies are required for the optimization of their usage for therapeutic purposes.

## 6. Aminopeptidase N (APN/CD13)

Out of the neo/angiogenesis-related molecules, another intensively investigated one is the aminopeptidase N receptor (APN/CD13) [148]. The association between Zn^2+^-dependent membrane-bound ANP/CD13 and cell differentiation, motility, proliferation, cell death, and angiogenic processes is well-established [149]. According to existing literature, among others, melanoma-, prostate-, pulmonary-, breast-, and ovarian-cancer cells express APN/CD13 [148,150,151,152]. Peptides with the asparagine–glycine–arginine sequence (tripeptide NGR) show selective APN/CD13pos. tumour-targeting capability [153]. Considering that APN/CD13 positivity is directly proportional to tumour aggressiveness, NGR-motif-based imaging probes could be promising in the identification of APN/CD13-overexpressing neoplasms. Beyond diagnostic applications, NGR-containing molecules also act as chemotherapeutic drug carriers, thus serving as a useful means of directed anti-cancer treatment choices [154,155,156]. Previous studies by Máté et al., Kis et al., and Gyuricza et al. have confirmed the diagnostic potential of NGR agents radiolabelled with ^68^Ga in the detection of tumours with APN/CD13 upregulation [157,158,159]. Other studies conducted by Li et al., Ma et al., and Vats et al. applied ^64^Cu, rhenium-188 (^188^Re), and ^177^Lu, respectively, for labelling purposes of NGR radiopharmaceuticals [160,161,162]. Although no former findings regarding ^44^Sc-labelled NGR-based radiotracers are available so far, ^44^Sc could be a potential radionuclide for the labelling of APN/CD13-targeting imaging vectors.

## 7. Conclusions

The in vivo tumour-targeting capability of ^44^Sc-labelled, angiogenesis-targeting imaging probes, along with the favourable physicochemical properties of the radiometal, capitalize on the relevance of this isotope for the development of highly specific PET radiotracers. Their transition into clinical routine will inevitably herald a new era in personalized cancer diagnostics. Furthermore, the non-invasive assessment of angiogenesis-/neo-angiogenesis-related molecules is of outstanding importance to identifying patients who may benefit from targeted antiangiogenic therapy. Together with its therapeutic surrogate, the ^44^Sc/^47^Sc isotopic pair could be effectively employed in dosimetry estimations, cancer treatment, and therapeutic follow-up.

However, the future progression of ^44^Sc-labelled molecular vectors from preclinical level to human application could be hampered by some constraints regarding the usage of ^44^Sc. First, the limited accessibility of the radiometal is a meaningful hurdle to its centralized distribution. In addition, the high-energy concomitant gamma rays (1157 keV) of ^44^Sc could lead to reduced image quality; therefore, future studies to optimize PET-imaging parameters are required [32]. Furthermore, ^43^Sc lacking in accompanying gamma co-emission may serve as a valuable substitute for ^44^Sc in imaging settings [46]. Global steps should be taken to overcome these drawbacks, with the ultimate goal of exploiting both the diagnostic and therapeutic value of ^44^Sc.

## Figures and Tables

**Figure 1 ijms-24-07400-f001:**
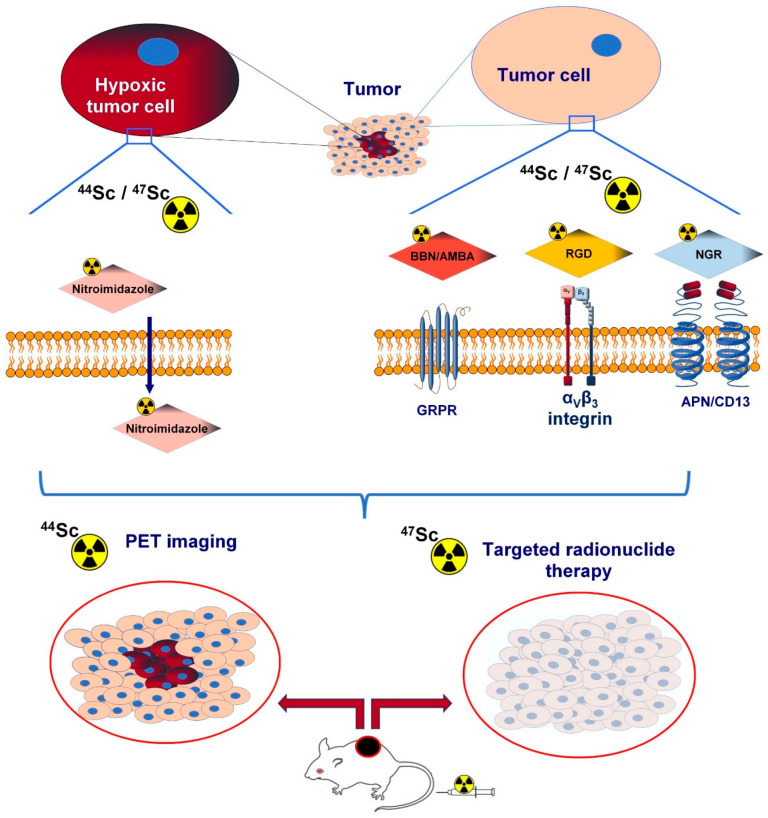
Sc^44^- and Sc^47^-labelled peptide-based radiopharmaceuticals in the molecular imaging and therapy of cancer-related angiogenesis.

**Table 1 ijms-24-07400-t001:** Preclinical studies with ^44^Sc-labelled PET radiotracers.

Investigated Object	Investigated Phenomenon	Target Molecule	(Radio) Labelled Vector	Imaging Technique	Reference
PCa PC-3 and HaCaT cell lines	In vitro receptor-binding affinity utilizing in vitro blocking studies with BBN	GRPR	[^44^Sc]Sc-NODAGA-AMBA and [^68^Ga]Ga-NODAGA-AMBA,	In vitro gamma counter measurements %ID/million cells units	[75]
PCa PC-3 tumor-bearing CB17 SCID mice and healthy control	In vitro and in vivo biodistribution pattern, tumor-targeting capability based on blocking experiments	GRPR	[^44^Sc]Sc-NODAGA-AMBA and [^68^Ga]Ga-NODAGA-AMBA	In vivo miniPET imaging, ex vivo radioactivity determination by gamma counter (%ID/g), in vivo and ex vivo blocking studies with BBN	[75]
PC-3 cell line	In vitro receptor-binding affinity with blocking studies applying [^125^I-Tyr^4^]-BN	GRPR	^nat^Sc-DOTA-BN[2-14]NH_2_ and ^nat^Ga-DOTA-BN[2-14]NH_2_	Competitive displacement cell-binding assay	[33]
Healthy male Sprague-Dawley rats, male Copenhagen rats bearing androgen-independent Dunning R-3327-AT-1 prostate cancer tumour	In vivo and ex vivo organ distribution, GRPR-targeting ability applying blocking studies with BBN	GRPR	^44^Sc-DOTA-BN[2-14]NH_2_ and ^68^Ga-DOTA-BN[2-14]NH_2_	In vivo dynamic microPET imaging (in tumourous rats), ex vivo (in healthy rats) radioactivity calculations with a dose calibrator (% ID/g)	[33]
U87MG and AR42J tumour-bearing female athymic nude mice (CD-1 nude)	Evaluation of in vitro and in vivo behaviour, in vivo and ex vivo biodistribution	α_v_β_3_ integrin	[^44^Sc]Sc-DOTA-RGD, [^44^Sc]Sc-NODAGA-RGD, [^68^Ga]Ga-DOTA-RGD, [^68^Ga]Ga-NODAGA-RGD	In vivo PET/CT acquisition, ex vivo gamma counting (%IA/g)	[76]
4T1 tumor-bearing BALB/c mice and healthy control	applicability of chelator AAZTA, in vivo biodistribution	α_v_β_3_ integrin	^44^Sc^3+^, and ^44^Sc(AAZTA)^−^ and ^44^Sc(CNAAZTA-c(RGDfK)	in vivo PET/MRI examinations	[77]
U87MG cells	In vitro receptor-binding affinity and specificity with blocking studies using ^125^I-Echistatin	α_v_β_3_ integrin	cRGD, (cRGD)_2_, and DOTA-(cRGD)_2_	In vitro competitive cell-binding assay, gamma counter-based detection of the tracer concentration	[17]
U87MG glioblastoma-bearing female athymic nude mice	Tumour-targeting competence, specificity applying in vivo and ex vivo receptor blocking with c(RGD)_2_, in vivo and ex vivo biodistribution	α_v_β_3_ integrin	[^44^Sc]Sc-DOTA-c(RGD)_2_	In vivo microPET/microCT imaging, ex vivo radioactivity determination (%ID/g)	[17]
CB17 SCID adult male mice bearing KB tumours and healthy control mice	Synthesis procedure, comparison of ^44^Sc- and ^68^Ga-labelled derivatives, in vivo and ex vivo biodistribution	Hypoxia	[^44^Sc]Sc-DO3AM-NI and [^68^Ga]Ga- DO3AM-NI	In vivo PET/MRI studies, ex vivo radiopharmaceutical uptake measurement %ID/g	[78]
Pro-angiogenic VEGF-A_165_/NRP-1 complex formation	Investigation of physicochemical properties and affinity for NRP-1	NRP-1	^44^Sc-radiocompounds (^44^Sc-1, ^44^Sc-1bis, ^44^Sc-2, ^44^Sc-3):Sc-DOTA-Ahx-A7R (Sc-1), Lys(DOTA-Sc)-A7R (Sc-1bis), Lys(hArg)-Dab(Ahx-DOTA-Sc)-Pro-Arg (Sc-2) and DR7A-DLys(DOTA-Sc) (Sc-3)	Competitive ELISA test	[79]

**Table 2 ijms-24-07400-t002:** Physical characteristics of commonly used PET radionuclides.

	^18^F	^68^Ga	^64^Cu	^43^Sc	^44^Sc	^89^Zr
Half-life (h)	1.83	1.13	12.7	3.89	3.97	78.41
Decay method (%)	EC (3) ß^+^ (97)	EC (11.1) ß^+^ (88.9)	EC (43.9) ß^+^ (17.6) ß^−^ (38.5)	EC (12) ß^+^ (88)	EC (5.7) ß^+^ (94.3)	EC (77.3) ß^+^ (22.7)
β^+^ endpoint energy, keV (%)	633.5 (96.7%)	1899 (87.7) 822 (1.2)	653 (17.6)	1200 (70.9) 826 (17.2)	1474 (94.3)	902 (22.8)
Principal γ energies, keV (Abs.%)	511 (194)	511 (177.8) 1077 (3.2) 1261 (0.1) 1883 (0.1)	511 (35.2) 1346 (0.5)	372.8 (23)	511 (188.5) 1157 (99.9) 1499 (0.9)	511 (45.5) 909 (99.0) 1713 (0.7) 1745 (0.1)

EC: electron capture.

## Data Availability

The datasets used and/or analysed during the current study are available from the corresponding author upon reasonable request.

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
