# Peer review of "Scandium-44: Diagnostic Feasibility in Tumor-Related Angiogenesis"

_ijms, 2023, doi:10.3390/ijms24087400_

Round 1

Reviewer 1 Report

The review article entitled “Scandium-44: diagnostic feasibility in oncological research” by Trencsenyi and Kepes is comprehensive review that compiled the applications of Scandium-44 in positron emission tomography (PET) imaging. Authors have compared half-life of Scandium-44 with other radioactive tracers and how it is beneficial than the recent novel radionuclides like Gallium-68 (68Ga) or Copper-64 (64Cu).   Authors justify and recommends the use the of Scandium as one of radionuclides used for the PET imaging. There are few articles in the field of oncology on Scandium-44 but not extensive as this one. The authors have done amazing job in compiling the article references according to clinical relevance in oncology research.

Interesting article by Lima et al, 2021 (PMID: 34679525) studied the PET capabilities using Sc-43 and Sc-44 and other conventional clinical radionuclides. It would be appropriate to include Lima,2021 as one of the references in the current review article. Another recent article by Ghiani S etal ,2021 (PMID: 33420915) show Scandium-44 conjugated to PMSA inhibitor for high efficiency prostate cancer imaging. Please include this reference in the current review article which overlaps with the interests of this article. Authors have discussed the short half-life and related transport challenges of 68Ga which is currently used in the PET imaging and how Scandium-44 could be a valuable substitute. Fig 1 very well depicts comprehensive overview about the role of 44Sc-labelled peptide-based radiopharmaceuticals in the molecular imaging of cancer-related angiogenesis. Authors discussed some of the important in vivo finding from others groups, including [44Sc]Sc-NODAGA-AMBA uptake of the PC-3 tumors were more prominent in comparison with the 68Ga-labelled derivative, the GRPR-selectivity of [44Sc]Sc-NODAGA-AMBA - along with the better imaging characteristics of 44Sc over 68Ga - 44Sc-labelled BBN based radiotracers. Authors claim that the similar uptake and excretion times of both the 68Ga and the 44Sc-labelled radiopharmaceuticals as well as their comparable accumulation in neoplastic tissues, would make these radiometals seemed equally effective in the detection of GRPRpos. tumors.

Conceptual comments

The manuscript is clear, comprehensive and of relevance to the oncology diagnostics. Important articles in the recent past have been included and justified in the manuscript. There are no excessive self-citations. In last 10 years there has not been good enough compilation of applications of Scandium-44 in diagnostic PET for oncology research. This information has been made available in the current manuscript. It is therefore relevant and a great addition to the existing knowledge of the scientific community. In my opinion adding a few of the recent the articles like PMID: 34679525 and PMID: 33420915would certainly strengthen the manuscript.

The conclusions and future perspectives are consistent with the evidence and arguments presented in the manuscript. Together with its therapeutic surrogate, the 44Sc/47Sc isotopic pair could be effectively employed in cancer treatment and therapeutic follow-up. Limited accessibility of radiometal, Sc-44 and high energy gamma rays and some of the drawbacks of Sc-44 has been mentioned in the conclusion.

In my opinion this manuscript will help to increase the knowledge in oncology research and add value in PET probes used in tumor diagnosis.

Reviewer 2 Report

General comments

PET imaging has been developing for decades now and radiopharmaceuticals with impressive performances in functional imaging have been synthesized. PET is now a routine clinical procedure. While F-18 is by far the most often used radionuclide, the interest for Ga-68, Zr-89 and Cu-64 is growing, with Ga-68 now included in clinical routine. Sc-44 has been proposed several years ago as another radionuclide of interest because of its physical properties, with an intermediate half-life of about 4 hours, favorable for later imaging, and efficient labeling using DOTA. A review of the results obtained with this radionuclide is definitely of great interest.

The review should more clearly define the advantages and drawbacks of Sc-44, compared to other radionuclides and particularly Ga-68 and Cu-64. It should help us answer the questions: is there room for Sc-64 in PET imaging and is it worth developing it as a routine radionuclide?

Answering the question for any possible application in oncology would possibly involve too many publications. Focusing on neo-angiogenesis would be a good idea. The number of publications is large enough but not overwhelming. If other examples, i.e., bombesin or nitroimidazole analogues brings useful additional information, then the section of these examples should be more clearly explained.

The review is reasonably exhaustive in this context, but it should state its goals more clearly, particularly in the abstract and in the title. "Scandium-44: diagnostic feasibility in oncological research" is too broad. Once the scope of the review is clearly defined, it should closely focus on it.

The review is hard to follow. Published results should be better organized to make the assessment of the interest of Sc-44 clearer, based on a limited series of examples.

The review should focus on the potential advantages and drawbacks of Sc-44 as compared to the other radionuclides such as Ga-68 and Cu-64, but also F-18, for a variety of targeting agents. This is not so easy indeed, but the review could be clarified, emphasizing cases where Sc-44 gives distinct results, possibly superior:

-          Sc-44 has a longer half-life than Ga-68. Does that give it a real imaging advantage?

-          It appears that several studies indicate lower liver accretion using Sc-44 as compared to Ga-68. Is there an explanation? Is that a general finding?

-          Even if Sc-44 proves better than other radionuclides, the advantages appear to be moderate. Then the production and logistics of Sc-44 should be discussed in more details. Why is Sc-44 coming so late in the competition? This kind of question should also be considered in the review.

Specific comments

The abstract is confusing. It does not clearly state which is the object of the review: the use of scandium-44 as a PET radionuclide, the use of small molecules to target cancer cells or the imaging of tumor-related angiogenesis.

The keyword "aminobenzoic-acid (AMBA)" must be corrected.

"new-born capillary blood vessels" does not seem right for angiogenic blood vessels and could be misleading.

"A myriad of PET radionuclides" is a gross exaggeration. Less than 10 radionuclides have been really considered for PET imaging and, by the way, indium-111 is not positron emitter. To help the reader, a table comparing the emission properties of the PET radionuclide of interest (including Sc-44, Sc-43, Ga-68, Cu-64, Zr-89 and F-18) should be added.

A radionuclide is not synthesized. A more appropriate wording should be used.

Producing Ti-44 is not particularly cumbersome. The problem here is the very long haff-life of Ti-64, which is an issue for radioactive waste handling.

Scandium-44 has indeed a high energy gamma emission, but the reference to Koumarianou is not appropriate.

Radionuclide emitting lower energy positron produce better quality images. For the comparison with gallium-68, the energy of gallium-68 positrons should also be given.

The possibility of using scandium-44 for "dosimetry investigations in theranostic settings" is indeed of interest. However, the therapeutic radionuclides that could benefit from the dosimetry assessment should be mentioned.

"the global distribution" of scandium-44 remains far from a reality.

In the reference by Nagy et al. there is no indication that "the pharmacokinetic properties of 44Sc-appended imaging probes surpass those of the Ga-98 labelled counterparts".

"myriad" again. Strictly speaking myriad refers to 10,000. Use more realistic qualifications.

The Sc-44/Sc-47 radionuclide pair would indeed be very interesting. However, Sc-47 availability is even more limited than that of Sc-44. Radionuclide currently used for therapy in human are Lu-177 and, for clinical trials, Ac-225. The possibility of using Sc-44 for dosimetry assessment with these radionuclides should be discussed.

"immense number of experiments", "Era before Scandium-44", there are outrageous overstatements. Table 1 summarizes less than a dozen of studies and not a single clinical trial.

The high energy photon of Sc-44 is indeed an issue with respect to radiation safety and possibly dosimetry. Zr-89 also emits a high energy photon and that limited the acceptable activity, but the half-life is much longer. A few words of discussion about Zr-89, which is of much broader use than Sc-44, would be welcome.

In addition, it is proposed to use the high energy photon to develop 3-photon imaging. This could be added as a potential advantage of Sc-44 over other positron-emitting radionuclides.

Round 2

Reviewer 2 Report

The authors have extensively edited the manuscript and focused it on imaging tumor vasculature using scandium-44, an interesting PET radionuclide that may be used more frequently in the future.

A confusion remains to be corrected in the key words and in the Bombesin section : in the context AMBA does not refer to aminobenzoyl acid, but to aminobenzoyl-bombesin analogue or to 4-aminobenzoyl-Q-W-A-V-G-H-L-M-NH(2).

Author Response

Dear Reviewer 2,

Thank you for your precious review and valuable comments made on our present manuscript. Please find our answers to your comment below. All changes made in the main text could be followed with track changes.

Comment:

The authors have extensively edited the manuscript and focused it on imaging tumor vasculature using scandium-44, an interesting PET radionuclide that may be used more frequently in the future.

A confusion remains to be corrected in the key words and in the Bombesin section: in the context AMBA does not refer to aminobenzoyl acid, but to aminobenzoyl-bombesin analogue or to 4-aminobenzoyl-Q-W-A-V-G-H-L-M-NH(2).

Response:

Based on the request of the Reviewer we corrected the confusion related to AMBA in the main text, in the keywords section and in the abbreviation of list of Figure 1 and Table 1 as well.

Keywords:

The term „aminobenzoyl acid (AMBA)” was corrected and the following was added: „aminobenzoyl-bombesin analogue (AMBA)”.

Abstract:

Previous studies dealt with the evaluation of 44Sc-appended avb3 integrin-affine Arg-Gly-Asp (RGD) tripeptides, GRPR selective bombesin analogue aminobenzoyl acid (AMBA), and hypoxia-associated nitroimidazole derivatives in the identification of various cancers using experimental tumor models.

This sentence was corrected to the following:

Previous studies dealt with the evaluation of 44Sc-appended avb3 integrin-affine Arg-Gly-Asp (RGD) tripeptides, GRPR selective aminobenzoyl-bombesin analogue (AMBA), and hypoxia-associated nitroimidazole derivatives in the identification of various cancers using experimental tumor models.

Bombesin section:

Aminobenzoyl acid (DO3A-CH2CO-G-[4-aminobenzoyl]-QWAVGHLM-NH2; AMBA) is regarded as a highly valuable imaging agent in the nuclear medical diagnostics of GRPR upregulated tumors (Lantry et al., 2006).

This sentence was corrected to the following:

Aminobenzoyl-bombesin analogue (4-aminobenzoyl-Q-W-A-V-G-H-L-M-NH2, AMBA) is regarded as a highly valuable imaging agent in the nuclear medical diagnostics of GRPR upregulated tumors (Lantry et al., 2006).

Furthermore, the targeting potential of DOTA-conjugated BBN-analogue AMBA (DO3A-CH2CO-G-(4-aminobenzoyl)-QWAVGHLM-NH2) labelled with 68Ga or 177Lu was also reported at preclinical level (Lantry et al., 2006, Schroeder et al., 2010).

This sentence was corrected to the following:

Furthermore, the targeting potential of DOTA-conjugated AMBA labelled with 68Ga or 177Lu was also reported at preclinical level (Lantry et al., 2006, Schroeder et al., 2010).

Trusting in your positive evaluation,

Yours sincerely,

Zita Képes
